# Kaempferol and Its Glycoside, Kaempferol 7-O-rhamnoside, Inhibit PD-1/PD-L1 Interaction In Vitro

**DOI:** 10.3390/ijms21093239

**Published:** 2020-05-03

**Authors:** Ji Hye Kim, Young Soo Kim, Jang-Gi Choi, Wei Li, Eun Jin Lee, Jin-Wan Park, Jaeyoung Song, Hwan-Suck Chung

**Affiliations:** 1Korean Medicine (KM)-Application Center, Korea Institute of Oriental Medicine (KIOM), Dong-gu, Daegu 41062, Korea; jkim2903@kiom.re.kr (J.H.K.); yskim527@kiom.re.kr (Y.S.K.); jang-gichoi@kiom.re.kr (J.-G.C.); liwei1986@kiom.re.kr (W.L.); dldmswls0416@naver.com (E.J.L.); 2New Drug Development Center, Daegu Gyeongbuk Medical Innovation Foundation (DGMIF), Dong-gu, Daegu 41061, Korea; jinwanpark@dgmif.re.kr (J.-W.P.); jysong@dgmif.re.kr (J.S.)

**Keywords:** programmed cell death protein 1, programmed death-ligand 1, kaempferol, kaempferol 7-O-rhamnoside, kaempferitrin, immune checkpoint inhibitor, small molecule inhibitors

## Abstract

Kaempferol (KO) and kaempferol 7-O-rhamnoside (KR) are natural products from various oriental herbs such as Geranii Herba. Previous studies have reported some biological activities of KO and KR; however, their effects on PD-1/PD-L1 interaction have not been reported yet. To elucidate their inhibitory activities on PD-1/PD-L1 protein–protein interaction (PPI), biochemical assays including competitive ELISA and biolayer interferometry (BLI) systems were performed. Cellular PD-1/PD-L1 blocking activity was measured in a co-culture system with PD-1 Jurkat and PD-L1/aAPC CHO-K1 cells by T-cell receptor (TCR) activation-induced nuclear factor of activated T cells (NFAT)-luciferase reporter assay. The detailed binding mode of action was simulated by an in silico docking study and pharmacophore analysis. Competitive ELISA revealed that KO and its glycoside KR significantly inhibited PD-1/PD-L1 interaction. Cellular PD-1/PD-L1 blocking activity was monitored by KO and KR at non-cytotoxic concentration. Surface plasmon resonance (SPR) and biolayer interferometry (BLI) analysis suggested the binding affinity and direct inhibition of KR against PD-1/PD-L1. An in silico docking simulation determined the detailed mode of binding of KR to PD-1/PD-L1. Collectively, these results suggest that KR could be developed as a potent small molecule inhibitor for PD-1/PD-L1 blockade.

## 1. Introduction

Programmed cell death protein 1 (PD-1) is one of the immune checkpoints and is known to be involved with immunological self-tolerance [1,2]. Various immune cells, including activated T cells, have been reported to express PD-1 [2,3]. Thus, PD-1 plays a suppressive role toward T cell activity in peripheral tissues and in the tumor microenvironment [4].

Cytokines from the tumor microenvironment such as interferon-gamma (IFN-γ) induce PD-1 ligand-1 (PD-L1), and the PD-1 ligands are found on most cancers. Interaction of PD-1 and PD-L1 is closely associated with impaired T cell functions [5], allowing cancer cells to escape immune surveillance [6,7]. Therefore, reversal of T cell dysfunction through blocking of PD-1/PD-L1 is considered an important strategy for enhancing immunity to cancer.

In recent decades, it has been thought that an ideal strategy for cancer immunotherapy would be an immune checkpoint blockade (ICB) targeting the PD-1/PD-L1 pathway [8]. Most of these inhibitors are monoclonal antibodies; however, some limitations have been reported, including high costs for patients, restricted tumor burden penetration and unresponsiveness of ICB therapy [9]. To overcome these shortcomings, small molecules with low molecular weight have gained interest across chemical-based, and peptide-based checkpoint inhibitors.

Natural substances have been reported to possess potential for diverse biological activities with various molecular mechanisms [10,11,12,13]. Notably, recent studies have been focused on modulators of the PD-1/PD-L1 pathway to suppress development and progression of cancers [14,15,16,17]. Recently, several studies have reported that phenolic compounds from *Glycyrrhiza uralensis* [18] and caffeoylquinic acid compounds [19] showed inhibitory activities on PD-1/PD-L1 protein–protein interaction (PPI) [18,19]. Therefore, traditional herbal medicinal resources have possessed extensive potential as immune checkpoint modulators for immunotherapeutic agents.

The present study found that Geranii Herba extract (GHE) is a novel candidate agent for PD-1/PD-L1 inhibition. GHE was reported to contain various phytochemicals including flavonoids and phenolic compounds [20,21]. Among them, kaempferitrin (KI, kaempferol-3,7-dirhamnoside) was identified as one of the abundant compounds of GHE in our previous reports [22]. Interestingly, KI has been known to be hydrolyzed to kaempferol (KO) and kaempferol 7-O-rhamnoside (KR) in the human intestine by the gut microbiome [23]. In addition, KO was generated by enzymatic hydrolysis with α-l-rhamnoside and/or β-glycosidase from KI and KR in vitro [24].

Previous studies on KO and KO rhamnosides have reported diverse biological activities, including anti-oxidant [25], anti-inflammatory [24], and anti-tumor activities [26]. Although they have been widely examined, their PD-1/PD-L1 blockade effects have not yet been studied; to the best of our knowledge, this study is the first to describe their potential for PD-1/PD-L1 inhibition.

## 2. Results

### 2.1. Effects of KO and Its Glycosides on PD-1/PD-L1 Protein Interaction

To elucidate a potent candidate agent as a PD-1/PD-L1 interaction inhibitor, the effect of GHE, which contains KO and its glycosides, KR and KI (Figure 1), was examined using a competitive ELISA according to a previous study [27]. As a positive control, PD-1 or PD-L1 neutralizing antibody (αPD-1 or αPD-L1) and small molecule PD-1/PD-L1 inhibitor C1 were used (Figure 2A–C). The result showed that GHE dose-dependently inhibited PD-1 and PD-L1 protein–protein interaction (PPI) at an IC_50_ value of 87.93 μg/mL (Appendix A). To determine which active compounds of GHE have inhibitory effects on PD-1/PD-L1 interaction, a comparison study was performed. As shown in Figure 2D, KO showed the best blocking effect with an IC_50_ of 7.797 μM. KR and KI also revealed inhibitory effects on PD-1/PD-L1 binding but did not show dose-dependent activities. These results indicated that the active compounds of GHE on PD-1/PD-L1 blockade may be KO and its glycosyl compounds.

### 2.2. Effects of KO and Its Glycosides on PD-1/PD-L1 Interaction in a Cell Model System

It has been widely reported that the PD-1/PD-L1 axis is closely related to T cell function, and the reversal of T cell dysfunction has been suggested as an effective immune therapeutic strategy against cancer [28,29,30]. To screen and evaluate inhibitors for the PD-1/PD-L1 blockade, the effects of KO and its glycosides were investigated using the PD-1/PD-L1 blockade bioassay system [31,32]. In this system, two cell model systems were utilized; immortalized human T lymphocyte cells (Jurkat cells) were altered to constitutively express PD-1 and a T-cell receptor (TCR)-inducible nuclear factor of activated T cells (NFAT)-luciferase reporter (PD-1 Jurkat T cells), and CHO-K1 cells were modified to stably express human PD-L1 and TCR agonist (PD-L1/aAPC CHO-K1 cells) for production of antigen-presenting surrogate CHO cells [8]. First, to exclude the cytotoxic effect of KO and its glycosides on each cell model system, a Cell Counting Kit-8 (CCK) assay was conducted (Appendix A). Results showed that all of the compounds (KO, KR, and KI) were not cytotoxic up to 100 μM in either cell line. Therefore, subsequent experiments were performed at the observed non-cytotoxic concentrations.

When co-cultured with PD-1 Jurkat T cells and PD-L1/aAPC CHO-K1 cells, TCR activation is restrained by PD-1/PD-L1 ligation and the NFAT-luciferase reporter activity. When PD-1/PD-L1 interaction is disrupted, TCR activation induces luminescence via activation of the NFAT pathway in co-culture systems [32]. Therefore, the effects of KO and its glycosides on PD-1/PD-L1 interaction in the cell model system were examined by measuring NFAT-luciferase reporter activity after co-culture with PD-1/NFAT Reporter-Jurkat cells and PD-L1/aAPC CHO-K1 cells. Results exhibited that KO and KR blocked PD-1/PD-L1 interaction via NFAT transcriptional activity in a dose-dependent manner (approximately 3 to 3.5 fold) (Figure 3A,B). There were no significant differences between KO and KR with EC_50_ values of 16.46 and 15.37 μM, respectively. On the other hand, KI had no effect at the indicated concentration (Figure 3C). These results confirmed the potential blocking ability of KO and KR as a small molecule inhibitor of PD-1/PD-L1 interaction in cell model systems, but not upregulation of PD-1 level (Appendix A). The results described above indicate that KO and its glycosides are involved with the PD-1/PD-L1 blockade, as confirmed by the biochemical assay.

### 2.3. The Binding Interaction and Affinity of KR with PD-1 and PD-L1

In order to measure the binding affinity of KR to PD-1 and PD-L1, BLI analysis was conducted in accordance with a previous study [33]. An increase of KR concentration revealed a wavelength shift in the BLItz sensorgram, indicating that KR was more bound to biotinylated PD-1 (Figure 4A) and PD-L1 (Figure 4B), respectively, when immobilized on the streptavidin (SA) sensor. The equilibrium dissociation constants (K_D_) of KR to PD-1 and PD-L1 were 3.11 × 10^−5^ and 1.97 × 10^−5^ M, with coefficients of determination, R^2^, of 0.9925 and 0.9958, respectively. Although KR displayed a slightly higher K_D_ value to PD-L1 than PD-1, it seems the affinities of KR to PD-1 and PD-L1 were comparable based on the results (Table 1). Additionally, to validate whether the molecular target of KR is PD-1 or PD-L1, SPR analysis was performed using Biacore T200 (GE Healthcare, Chicago, IL, USA) equipment according to a previous study [19]. Results showed that KR binds with recombinant PD-1 with a K_D_ value of 1.56 × 10^−4^ M (Appendix A); however, the K_D_ value of KR with PD-L1 was not determined. Unexpectedly, KO was not analyzed in the BLItz and SPR analyses; this may be due to its low water solubility. Overall, these results suggested that KR possessed the potential to block activity against the PD-1/PD-L1 interaction by directly targeting PD-1 and PD-L1.

### 2.4. KO and Its Glycosides Inhibit PD-1/PD-L1 Protein Interaction In Silico

To elucidate the detailed binding modes of KO and its glycosides with PD-1/PD-L1, binding energies obtained from molecular docking simulation were elucidated by in silico modeling (Figure 5). KO and its glycosides attached to the PD-L1 region at the site of PD-1, but the detailed mode of action was different. The predicted binding energy of PD-1 to KO was −5.4 kcal/mol, followed by −5.6 kcal/mol of KR (Figure 5A, left panel). In the case of PD-L1, it was KO for −5.0 kcal/mol and KR for −5.3 kcal/mol, respectively (Figure 5A, right panel). In addition, all KO and its glycosides showed a slightly lower or similar binding affinity to that of PD-1/PD-L1 inhibitor C1 (Figure 5A and Appendix A). These results theoretically confirmed that KR has a higher possibility of binding with PD-l and/or PD-L1 compared to KO.

According to a previous study [34], the PD-1/PD-L1 complex is generated via major interaction with critical residues of PD-1 (Val64, Ile126, Leu128, Ala132, Ile134) and PD-L1 (Ile54, Tyr56, Met115, Ala121, Tyr123). Based on this information, pharmacophore analysis was represented as a 2D interaction diagram of PD1 or PD-L1 with each compounds (Figure 5B). Carbon at the 7 position in the glycosyl group of KR interacted with hydrophobic residues of Gly124, Ile126, and Ile134 (red crown mark) at the PD-L1 binding site of PD-1 protein (Appendix A). Additionally, the glycoside group generated a hydrogen bond with residues Tyr68, Thr76, and Glu136 (green dashed line) at the PD-L1 binding site of the PD-1 protein. The hydroxyl group in positions at carbon 18 and 20 formed the interaction with residue Val64 (red crown mark), and that at carbon 15 interacted with Lue128 at the PD-L1 binding site of PD-1 protein. Therefore, major amino acids of KR for binding to PD-1 were identified as four residues: Val64, Ile126, Leu128, and Ile134. Collectively, we speculated that the glycoside group might be closely associated with the functional activity of KR in the PD-1/PD-L1 interaction.

## 3. Discussion

In recent decades, extensive developments have been achieved to develop cancer agents for cancer immunotherapy. Studies to develop small molecule inhibitors for relatively inexpensive cancer therapeutics are in progress [2]. CA-170 (also known as AUPM170 or PD-1-IN-1) and Aurigene-1 have been reported as potent small molecule PD-L1 antagonists; however, biochemical results suggested that they did not target hPD-L1 directly but may be mediated by another mode of action [35,36]. These cases emphasize the importance of detailed mechanisms based on evaluation methods from biochemical assay to in vivo assay. Therefore, careful development of small molecules for targeting the PD-1/PD-L1 immune checkpoint with the proper molecular mode of action is required.

Based on previous studies, KO and its glycosides have been reported to possess diverse pharmacological activities such as anti-oxidant [25], anti-inflammatory [37], and anti-virus activities [22]. To the best of our knowledge, the present study initially reported that KO and KR possess PD-1/PD-L1 inhibitory activity in vitro and in silico. KI was the most abundant component of GHE; however, only hydrolysate forms of KI (KO and KR) were effective in this study. These results suggest that KO and KR, as bioactive compounds of GHE, interfere with PD-1/PD-L1 interaction.

In silico modeling and pharmacophore assays were performed to elucidate the more detailed binding mode of KR and the PD-1/PD-L1 complex (Figure 5). Using the AutoDock Vina program, a docking study calculated the binding affinity of the PD-1/PD-L1 complex (Figure 5, Appendix A). These results were experimentally confirmed using BLI analysis (Figure 4), and the binding affinities of KR toward hPD-1 and hPD-L1 were confirmed as 3.11 × 10^−5^ and 1.97 × 10^−5^ M, respectively. In a previous study, Mazewski et al. reported that some small molecule inhibitors of PD-1/PD-L1 developed by Bristol-Myers Squibb (BMS) Company showed high K_d_ values ranging from μM to mM [38].

In the case of KO, similar to the previous report [38], KO could not be analyzed because of the limits of solubility and fastening of the peak. It was reported that KO and KR are slightly soluble in water, with a solubility of 440 mg/L (1.537 mM) for KO and 1.16 mg/mL (2.68 mM) for KR. Most of the kaempferol in plants contains glycoside moieties rather than in the free form, and it has been reported to affect bioavailability and bioactivity [39]. A previous study reported that KI is known to be hydrolyzed to KO and KR in the human intestine by the gut microbiome [40]. Interestingly, the present study confirmed that KI derivatives (KO and KR) generated by deglycosylation showed more potent PD-1/PD-L1 inhibitory activity in vitro (Figure 2, Figure 3 and Figure 4) and in silico (Figure 5), indicating potential for PD-1/PD-L1 interaction inhibitors. However, considering that KO reaches 35 nM to 1.61 μM concentration in rat and human plasma [41,42], it is required to modify KO to more active compounds by chemical or enzymatic transformations in order to enhance its water solubility and biological activity. Further studies on the bioavailability of KO and KR are required, and their in vivo efficacy studies would be needed to evaluate their potential PD-1/PD-L1 blocking effects in animal and human models.

## 4. Materials and Methods

### 4.1. Chemicals and Antibodies

RPMI 1640 medium (#SH30027.01), Dulbecco’s Modified Eagle’s Medium (DMEM, #SH30243.01), F-12 Kaighn’s Modification (#SH30526.01) medium, fetal bovine serum (FBS, #SH30084.3), 0.25% trypsin-EDTA, and penicillin-streptomycin (#SV30010) were purchased from Hyclone (South Logan, UT, USA). The human PD-1/PD-L1 Inhibitor Screening Assay Kit (#72005), human PD-1 (hPD-1) neutralizing antibody (#71120), and human PD-L1 (hPD-L1) neutralizing antibody (#71213) were purchased from BPS Bioscience (San Diego, CA, USA). Kaempferitrin (KI), kaempferol (KO), and kaempferol 7-O-rhamnoside (KR) were purchased from ChemFaces (Wuhan, Hubei, China). PD-1/PD-L1 Inhibitor C1 (M60312-2s) was purchased from Xcess Biosciences Inc. (San Diego, CA, USA). The PD-1/PD-L1 Blockade Bioassay Kit was purchased from Promega (Fitchburg, WI, USA).

### 4.2. Preparation of GHE

Extracts of Geranii Herba (GHE) was obtained from the NIKOM (National Development Institute of Korean Medicine, Gyeongsan, Korea). GHE were freeze-dried to powder and then dissolved in DMSO with gentle shaking. After centrifugation at 480× *g* for 20 min, the insoluble residues were removed and the supernatant was stored in a desiccator at 4 °C until the experiments [43].

### 4.3. Competitive Enzyme-Linked Immunosorbent Assay (ELISA)

To elucidate the effect of test inhibitors on PD-1/PD-L1 protein–protein interaction, PD-1/PD-L1 Competitive ELISA Screening Kits (BPS Bioscience Inc., San Diego, CA, USA) were used according to the supplier’s instructions [27]. Briefly, recombinant hPD-L1 (BPS Bioscience, #71104) was coated overnight at 1 μg/mL in phosphate-buffered saline (PBS) in 96-well plates (Corning Inc., New York, NY, USA). Plates were washed with PBS containing 0.1% Tween (PBS-T), blocked for 1 h at room temperature with PBS containing 2% (*w*/*v*) BSA, and then washed again. Fifty microliters of 0.5 μg/mL biotinylated hPD-1 (BPS Bioscience, #71109) was added to the wells, and the plates were incubated for 2 h at room temperature. After three washes in PBS-T, 50 μL of 0.2 μg/mL HRP-conjugated streptavidin was added to each well, and the plates were incubated for 1h. After incubation, plates were washed three times in 0.1% PBS-T, and relative chemiluminescence was measured on a SpectraMax L Luminometer from Molecular Devices (San Jose, CA, USA).

### 4.4. Cell Culture

Jurkat T cells expressing firefly luciferase gene under the control of NFAT response elements with constitutive expression of human PD-1 (PD-1 Jurkat cells, effector cells) were obtained from BPS Bioscience (San Diego, CA, USA). PD-1 Jurkat cells were cultured in RPMI 1640 medium (Hycolone) supplemented with 10% FBS, 1% penicillin, and streptomycin. CHO-K1 cells constitutively expressing human PD-L1 and an engineered T cell receptor (TCR) activator (PD-L1/aAPC CHO-K1 cells, target cells) were obtained from BPS Bioscience (San Diego, CA, USA). PD-L1/aAPC CHO-K1 cells were incubated in F12/DMEM containing 10% (*v*/*v*) heat-inactivated FBS and 1% penicillin/streptomycin at 37 °C and 5% CO_2_. Additionally, both cells were maintained in complete medium with Hygromycin B (200 μg/mL) and G418 (1 mg/mL) to select stable cell lines expressing the genetic constructs. For the experiments, Hygromycin B and G418 were not included in the medium.

### 4.5. PD-1/PD-L1 Blockade Assay

A PD-1/PD-L1 Blockade Bioassay kit was used according to the supplier’s instructions (Promega, Fitchburg, WI, USA) with a slight modification [8]. Briefly, the PD-L1 aAPC/CHO-K1 cells were seeded in 96-well plates at a density of 5 × 10^4^/well in the complete culture medium. A few hours later, the medium was removed, and 1 × 10^5^ PD-1 Jurkat cells were added to each well with test inhibitors. The cells were cultured for 24 h at 37 °C and then lysed using the Bio-Glo™ Luciferase Assay System (Promega, Madison, WI, USA). After incubation for 15 min, luminescence was quantified using a GloMax^®^ Explorer Multimode Microplate Reader (Promega). Data are presented as fold to untreated control.

### 4.6. Kinetic Analysis by Biolayer Interferometry (BLI Analysis)

To measure the binding affinities and kinetic constants of test inhibitors, BLI analysis was conducted according to a previous report, with slight modifications [33]. Briefly, BLI on a BLItz system is a label-free analytic technology for biomolecular interactions by measuring interference patterns of light reflected from two surfaces [32]. After pre-equilibration of the SA BLI sensor (Pall FortéBio Corp., Menlo Park, CA, USA) in PBS buffer for 10 min, biotinylated human PD-1 (BioVision, Milpitas, CA, USA) and PD-L1 (Sino biological, Beijing, China) were fully loaded onto the sensors by immersion in 4 μL of PD-1 and PD-L1 solution and then dissolved in PBS buffer to 50 μg/mL. Test inhibitors for the kinetic analysis were prepared by 100-fold dilution of stock solution and dissolved in 100% DMSO (0, 0.25, 0.5, and 0.75 mM) in PBS buffer. Binding kinetics were measured as follows: step 1, initial baseline in PBS buffer containing 1% DMSO for 15 s; step 2, association in 4 μL of test inhibitors solution for 20 s; and step 3, dissociation in PBS buffer containing 1% DMSO for 20 s. The kinetic constants were calculated using the BLItz Pro software by fitting the association and dissociation data to a 1:1 model. The equilibrium dissociation constant, K_D_, was calculated as the dissociation constant (k_d_)/association constant (k_a_).

### 4.7. In Silico Docking Simulation and Interaction Analysis

The test inhibitors were docked onto the interaction space between PD-1 and PD-L1 (PDB code: 4ZQK) retrieved from the Protein Data Bank (www.rcsb.org) and a previous report [34] using AutoDock Vina integrated with UCSF Chimera-alpha v1.13 [44]. The hydrophobic and hydrogen-bonding interactions between PD-L1 and each small molecule were analyzed using LigPlot+ v1.4.5 [45]. Amino acid residues involved in the interactions are indicated in red (hydrophobic interactions) and green (H-bonds).

### 4.8. Statistical Analysis

The data were expressed as mean ± the standard error (S.E.) of the mean. Differences in the mean values between the treatment and control groups were analyzed by one-way analysis of variance with Dunnett’s post-hoc test for multiple comparisons. GraphPad PRISM software^®^ Version 5.02 (La Jolla, CA, USA) was used for analysis. *P*-values less than 0.05 were considered significant. Statistical differences are indicated using asterisks.

## 5. Conclusions

The present study indicated that KO and KR are active small molecule inhibitors against PD-1/PD-L1 interaction. Biochemical and in silico assays confirmed that KR may be a potential PD-1/PD-L1 inhibitor targeting both PD-1 and PD-L1. Therefore, we suggest that this PD-1/PD-L1 blockade of KR can be explained by its ability to interfere with the PD-1 and PD-L1 binding sites of their complex. Further studies should investigate whether KO and KR can act as PD-1/PD-L1 inhibitors in vivo in order to develop them as potent pharmaceuticals for immune checkpoint blockade therapy.

## Figures and Tables

**Figure 1 ijms-21-03239-f001:**
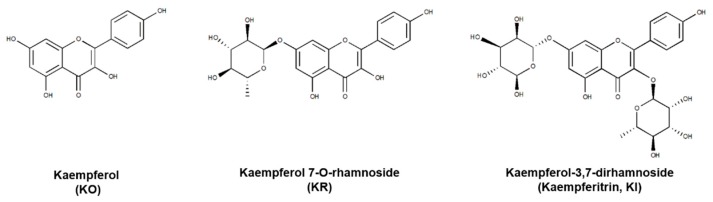
The chemical structures of kaempferol (KO), kaempferol 7-O-rhamnoside (KR), and kaempferol-3,7-dirhamnoside (kaempferitrin, KI). Chemical structures were generated using ChemDraw Professional 8.0.

**Figure 2 ijms-21-03239-f002:**
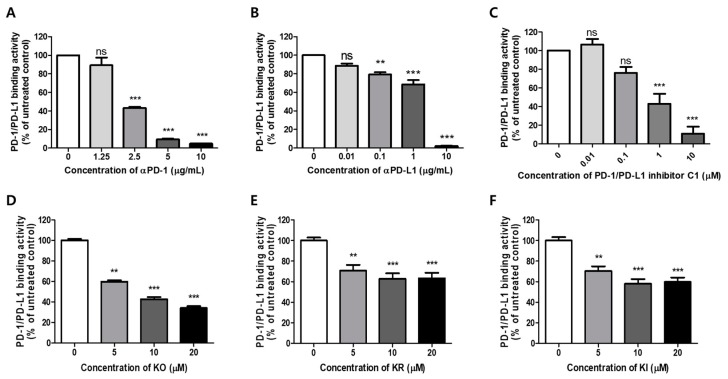
Effects of KO and its glycosides on programmed cell death protein 1 (PD-1)/PD-1 ligand-1 (PD-L1) protein interaction in a competitive ELISA. (**A**) PD-1 neutralizing antibody, (**B**) PD-L1 neutralizing antibody, (**C**) PD-1/PD-L1 inhibitor C1, (**D**) KO, (**E**) KR, and (**F**) KI were pre-treated onto plates coated with PD-L1, followed by incubation with biotinylated PD-1. Relative PD-1/PD-L1 binding activities were determined using a competitive ELISA assay, as described in the Materials and Methods. Data are presented as means ± S.E. (standard error) values of three independent experiments. Asterisks indicate significant inhibition of PD-1/PD-L1 binding activity by each test inhibitor as compared with the control group. (** *p*  <  0.01, and *** *p*  <  0.001; ns: not significant).

**Figure 3 ijms-21-03239-f003:**
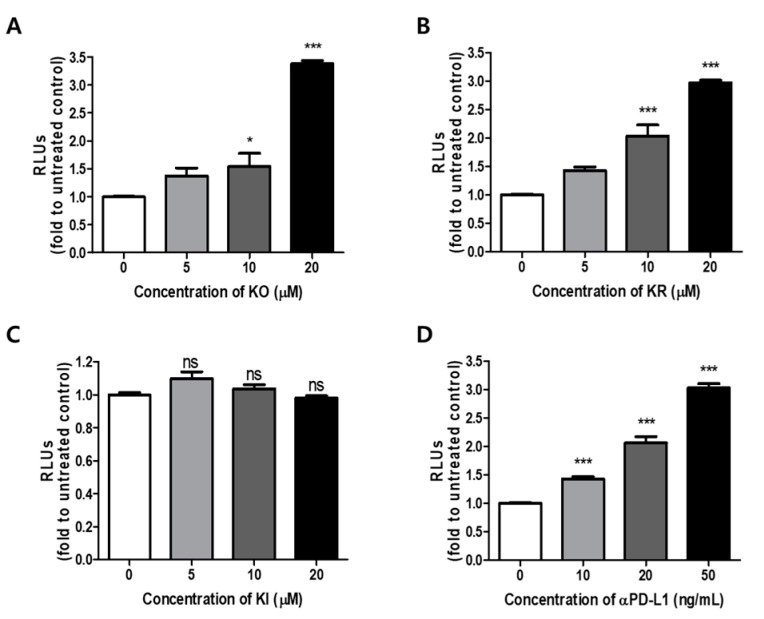
Effects of KO and its glycosides in cell-based PD-1/PD-L1 blockade assay. Effect of (**A**) KO, (**B**) KR, (**C**) KI and (**D**) aPD-L1 on T-cell receptor (TCR)-mediated nuclear factor of activated T cells (NFAT) activity by the PD-1/PD-L1 blockade in a co-cultured system. PD-1 Jurkat effector cells and PD-L1/aAPC CHO-K1 target cells were co-cultured with each compound for 24 h at the indicated concentrations. The activation of PD-1 Jurkat cells led to the level of NFAT-luciferase (for details, see Materials and Methods). Data are presented as means ± S.E. (standard error) values of three representative independent experiments. Asterisks indicate significant upregulation of relative luminescence units (RLUs) by each sample as compared with the control group (* *p*  <  0.05 and *** *p*  <  0.001; ns: not significant).

**Figure 4 ijms-21-03239-f004:**
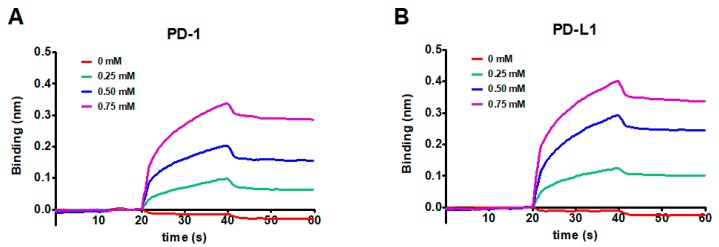
Global kinetic analysis of KR binding to biotinylated PD-1 (**A**) and PD-L1 (**B**) immobilized on a streptavidin biolayer interferometry (BLI) sensor. The binding of test inhibitors with PD-1 was confirmed by BLI analysis. The kinetics of KR for immobilized PD-1 and PD-L1 were monitored with increasing concentrations of KR (0, 0.25, 0.5, and 0.75 mM) dissolved in phosphate-buffered saline (PBS) buffer (pH 7.3) containing 1% DMSO. Data are presented as representative of three independent experiments.

**Figure 5 ijms-21-03239-f005:**
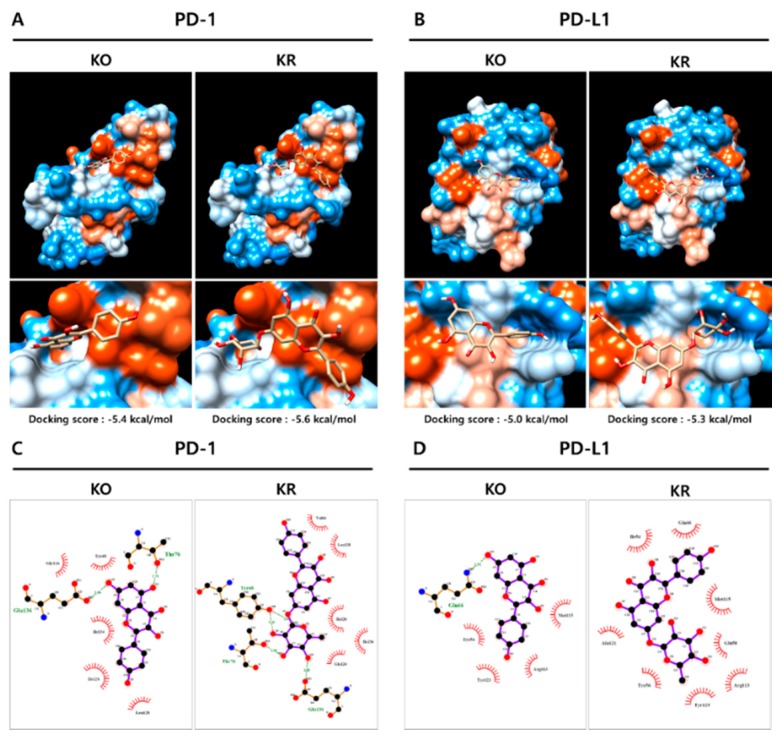
Hypothetical model of human PD-L1 or PD-1 in complex with KR. (**A**,**B**) Protein–ligand docking simulation for PD-1 (**A**) or PD-L1 (**B**) with each compound. Binding models were obtained through docking simulation derived from the PD-1/PD-L1 complex (PDB code 4ZQK) using AutoDock Vina. (**C**,**D**) Pharmacophore analysis of PD-1 (**C**) or PD-L1 (**D**). The hydrogen bonds and hydrophobic interactions between PD-L1/PD-1 and test inhibitors were analyzed using the LigPlot+ program.

**Table 1 ijms-21-03239-t001:** The kinetic analysis of KR binding to PD-1 and PD-L1 using BLItz system.

Protein	*K*_D_ (M)	*k*_a_ (M^−1^ s^−1^)	*k*_d_ (s^−1^)	*R* ^2^
PD-1	3.11 × 10^−5^	3.15 × 10^2^	9.79 × 10^−3^	0.9925
PD-L1	1.97 × 10^−5^	3.92 × 10^2^	7.73 × 10^−3^	0.9958

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
