# Peer review of "Kaempferol and Its Glycoside, Kaempferol 7-O-rhamnoside, Inhibit PD-1/PD-L1 Interaction In Vitro"

_ijms, 2020, doi:10.3390/ijms21093239_

Round 1

Reviewer 1 Report

The manuscript  ijms-781174, is thatnatural product, kaempferol (KO) andkaempferol-7-O-rhamnose(KR) isolated from various natural source, inhibits PD-1/PD-L1 Interaction In Vitro study, including the in silico study docking simulation determined the detailed mode of binding of KR to PD-1/PD-L1. This article is interesting in the fact that author investigated important  biological  activity  in detail such as that cellular PD-1/PD-L1 blocking activity was measured in a co-culture system with PD-1 Jurkat and PD-L1/aAPC CHO-K1 cells by TCR-induced NFAT-luciferase reporter assay. Accordingly, this manuscript may be accepted in Int. J. Mol. Sci.  after minor revision as below. Minor point 1) In Figure 1, the direction of “ bold” bond at anomeric position of KR and KI, is opposite, please correct it.  2) The reviewer considers that an information of contents in natural medicine such as GHE play an important roll in this type of study. How is contents of KO, LR and KI in GHE? When GHE can inhibits PD-1/PDL-1 in 50 micro g/L, is contents of KO, LR and KI 20%?(10 micro g/L of KO, LR and KI inhibit PD-1/PD-1L respectively. )

Author Response

27-Apr-2020

Dear Reviewer 1

Re: IJMS-781174
Kaempferol and its Glycoside, Kaemperol-7-O-rhamnose, Inhibits PD-1/PD-L1 Interaction In Vitro.

Ji Hye Kim, Young Soo Kim, Jang-Gi Choi, Wei Li, Eun Jin Lee, Jin-Wan Park, Jaeyoung Song and Hwan-Suck Chung,*

We wish to thank you and the reviewers for the thorough and constructive comments. We have carefully considered all feedback received and have addressed each comment in detail below.

Kind regards,

Jihye Kim, Ph.D.
Korea Institute of Oriental Medicine (KIOM)
Tel: +82-53-940-3840
Fax: +82-53-940-3899
E-mail: jkim2903@kiom.re.kr

Reviewer 2 Report

The strength of the submitted article is

  1. Topical issue – monoclonal antibodies both against PD-1 or PD-L1 have been largely tested for several tumours, some of them are registered and represent a real progress in anticancer therapy, so a small inhibitor of one of these could enrich the current list of possible novel drugs
  2. Combination of experimental data with in silico docking simulation

There are also several weak points:

  1. Kaempferol is not being absorbed after oral way, or do authors have a proof that it reaches micromolar concentration in plasma?
  2. Their data on transcription factor NFAT are rather contradictory to their background (e.g. p.1 ….NFAT increases expression of PD-1) or current knowledge „a NFAT-specific inhibitor led to a sharp reduction in PD-1 expression“ in Oestreich KJ, Yoon H, Ahmed R, Boss JM. NFATc1 regulates PD-1 expression upon T cell activation. J Immunol. 2008) and their results (p.4 „KO and KR induced NFAT transcriptional activity in a dose dependent manner  (approximately 3 to 3.5 fold)“ – so if the results of authors are true, kaempferol can decrease PD-1 and PD-L1 interaction but can also induce PD-1 expression, which logically will weaken the final effect
  3. Language needs significant improvement (see few examples below)
  4. Article also contains significant number of very unprecise, speculative or even misleading expressions which must be removed – see also few examples below

Minor comments

– chemicals should not start with upper case letters (e.g. in the abstract „and Kaempferol-7-O-rhamnoside“, r.39 „as Interferon…“)

r.37/38: „PD-1 expression in naive T cells is transiently expressed by NFAT after T cell receptor (TCR) activation“ – reformulation needed

r. 47 – „including the route of administration (intravenous injection) and restricted tumor penetration“ - monoclonal antibodies in cancer can be now also given subcutaneously, monoclonal antibodies does not penetrate the cells or are immediatelly cleft in lysosomes after endocytosis, so there is no real penetration inside the cells

r.50 – „Natural substances possesses some advantages including safety“ – non-sense, both synthetic and natural compounds can be very toxic, e.g. aconitine is a natural compound but it is very toxic in mg, but paracetamol is safe in few grams

Figure 3 should be shifter to supplementary data

Figure 4 – „Data are presented as means ± S.E. (standard error) values „ – I suppose that SEM are used as was specified in the statistical analysis

Data in the Table 1, Figure 5 and in the text do not correspond, the concentrations are in mM (0.25-0.75 mM) in the Figure 5 while in the text (r. 169) in microM (0.125-0.375 microM).

  1. 151 – „As a higher concentration of KR was treated“ ?
  2. 206 – „there are extensive developments have been achieved“

r.208-212 – „Guangzhou Maxinovel Pharmaceuticals announced MAX-10129 which revealed enhanced oral bioavailability and anti-tumor efficacy in MC38 tumor model [2]. It is currently being examined that triple and quadruple combined therapies with MAX-10129 an anti-CTLA4 antibody, Epacadostat (IDO inhibitor), Celebrex (COX-2 inhibitor), and cisplatin (anti-cancer agent) [2]. The diverse peptide-based inhibitors such as BMS-986189, CA-170 and CA-327 have been in progress for clinical trials [2].“ – clear relationship to this article should be revealed in this part, anyway, this part can be shortened

  1. 215 – „suggested some assumption“
  2. 223- „Although previous study report“

r.231 „BMS“ – what is BMS ?

r.265 – rpm should be expressed in g

chapter 4.5 – the principle of the method should be at least shortly mentioned

chapter 4.7 – BLI and Blitz systems should be also at least briefly explained

Author Response

27-Apr-2020

Dear Reviewer 2

Re: IJMS-781174
Kaempferol and its Glycoside, Kaemperol-7-O-rhamnose, Inhibits PD-1/PD-L1 Interaction In Vitro.

Ji Hye Kim, Young Soo Kim, Jang-Gi Choi, Wei Li, Eun Jin Lee, Jin-Wan Park, Jaeyoung Song and Hwan-Suck Chung,*

We wish to thank you and the reviewers for the thorough and constructive comments. We have carefully considered all feedback received and have addressed each comment in detail and please see the attachment.

Kind regards,

Jihye Kim, Ph.D.
Korea Institute of Oriental Medicine (KIOM)
Tel: +82-53-940-3840
Fax: +82-53-940-3899
E-mail: jkim2903@kiom.re.kr

Round 2

Reviewer 2 Report

authors reacted on my all coments, the article is now acceptable from my point of view